# Molecular Mechanism of Conformational Crossover of Mefenamic Acid Molecules in scCO_2_

**DOI:** 10.3390/ma16041403

**Published:** 2023-02-07

**Authors:** Roman D. Oparin, Mikhail A. Krestyaninov, Dmitry V. Ivlev, Michael G. Kiselev

**Affiliations:** G.A. Krestov Institute of Solution Chemistry of the Russian Academy of Sciences (RAS), Akademicheskaya St. 1, Ivanovo 153045, Russia

**Keywords:** mefenamic acid, supercritical carbon dioxide, true solution, conformational equilibrium

## Abstract

In this work, we studied conformational equilibria of molecules of mefenamic acid in its diluted solution in scCO_2_ under isochoric heating conditions in the temperature range of 140–210 °C along the isochore corresponding to the scCO_2_ density of 1.1 of its critical value. This phase diagram range totally covers the region of conformational transitions of molecules of mefenamic acid in its saturated solution in scCO_2_. We found that in the considered phase diagram region, the equilibrium of two conformers is realized in this solution. In the temperature range of 140–180 °C, conformer I related to the first, most stable polymorph of mefenamic acid prevails. In the temperature range of 200–210 °C, conformer II, which is related to the second metastable polymorph becomes dominant. Based on the results of quantum chemical calculations and experimental IR spectroscopy data on the mefenamic acid conformer populations, we classified this temperature-induced conformational crossover as an entropy-driven phenomenon.

## 1. Introduction

Conformational equilibria of active pharmaceutical ingredients (API) attract both academic and practical interest. This interest is mainly associated with the polymorphism of crystalline API [1,2], which is of great importance to pharmacy [3,4], because polymorphic forms of the same drug compound demonstrate dissimilar physicochemical properties that, in turn, predetermine different pharmacological properties. Currently, special attention is being paid to studying polymorphism and conformational equilibria of APIs in supercritical fluids (SCF) (see, e.g., [5]), when the role of the solvent is played by supercritical carbon dioxide (scCO_2_).

In a series of our recent works [6,7,8,9,10,11,12] we have shown that it is possible to control polymorphism of micronized forms of drugs, which are obtained using SCF technologies, by monitoring the conformer distribution of these drug molecules in an SCF solution by varying its parameters of state (P, T). In these works, where we used scCO_2_ as a solvent, we showed that when there is an equilibrium between the drug compound and the scCO_2_ phase, a conformational equilibrium of the drug molecules is formed in this fluid phase. This equilibrium is determined either by the polymorphic state of the crystalline API [6,8,9,10,11] or by the conformational equilibrium of its molecules when the API is in its amorphous form [7,12].

Mefenamic acid (MA) is a representative of the conformational type of polymorphism. It can exist in three polymorphic forms [13,14,15,16,17,18]. However, only two of them (I and II) are more stable [14]. As it follows from [14], under ambient conditions, the stability of MA polymorphs decreases as follows: I > II > III; at higher temperatures, an enantiotropic transformation changes this order to II > I > III. However, form III is not stable and cannot be obtained by direct crystallization [14]. Thus, in our previous work [8] we neglected the formation of this polymorph. For the same reason in the present study, we considered the conformers related to polymorphs I and II only.

According to the definition of conformational polymorphism, each of these polymorphic forms is determined by the corresponding MA molecule conformer. In our previous work [8], we showed that, along with the first stable polymorphic form, the commercial MA substance can contain an admixture of the second, less stable one (~13.6%). This fact is also confirmed by the results of [18,19], where the authors mentioned that the percentage of the second MA polymorph in the commercial substance can reach 10%. Thus, in the low-temperature range studied, we found that the saturated solution phase of the heterogeneous MA–scCO_2_ mixture contained conformer II, which corresponds to the second polymorph and conformer I related to the prevailing first polymorph [8]. Moreover, we found that the population of conformer II correlates with the mole fraction of polymorph II that exists in the bottom phase. The authors of [20,21] also mentioned that adding a small amount of a co-solvent (DMSO) to heterogeneous mixtures of mefenamic or tolphenamic acid with scCO_2_ (used to increase the acids solubility in the scCO_2_ phase) does not change this conformational equilibrium. In work [8], we also demonstrated that a temperature-induced change in the polymorphic composition of the bottom phase of a heterogeneous MA–scCO_2_ mixture shifts the conformational equilibrium of MA molecules in the fluid phase of this mixture. Finally, a full temperature-induced polymorphic transition of the crystalline MA results in degeneration of conformer I within the scCO_2_ phase. In its turn, the polymorphic conversion is related to the stability of MA forms in different phase diagram regions (see above).

It is obvious that in the case of a heterogeneous MA–scCO_2_ mixture, it is important to take into account the solid/fluid equilibrium when considering the conformational equilibrium in the saturated solution. Indeed, continuous exchange between the solid phase and solution must yield a conformer related to the most stable polymorph in the bottom phase of the system. However, the conformational transition can occur due to the change in the conformer free energies following variations in the parameters of state. In our previous work [8] we showed that the conformational transitions of the MA molecules in a saturated MA solution in scCO_2_ occur in the temperature range of 160–190 °C, whereas below 160 °C the conformational equilibrium remains constant. This fact predetermined the choice of the phase diagram region (temperature range of 140−210 °C) to be studied in the present work. Here we also worked under isochoric conditions (along the isochore corresponding to the scCO_2_ density equal to 1.1 of its critical value) in order to exclude the medium density effect. This study allowed us to understand what the driving force of conformational transitions of MA molecules in its SCF solution can be because the results of our previous works [6,7,8,9,10,11,12] did not clarify the situation.

Thus, the aim of the present work was to study the molecular mechanism of conformational transitions of MA molecules in an MA true solution in scCO_2_. To achieve this goal, the following tasks were formulated:within the framework of quantum chemical calculations, using different functionals, to carry out conformational search for MA molecules, to determine the energy characteristics of the conformers and vibration frequencies of the key functional groups involved in conformational transitions, and to determine the parameters of probable intramolecular H-bonds in different conformers of MA molecules;within the framework of molecular dynamics simulation, using different force fields, to determine the effect of the medium (scCO_2_) and temperature on the energy characteristics and probability of formation of different MA conformers;using in situ IR spectroscopy, to conduct a detailed study of the conformational equilibrium of MA molecules in a true solution in scCO_2_ under isochoric heating conditions in the temperature range of 140−210 °C (along the isochore corresponding to the scCO_2_ density equal to 1.1 of its critical value);to compare the results obtained for the true solution with those obtained in [8] for a heterogeneous MA–scCO_2_ mixture.

## 2. Materials and Methods

### 2.1. Materials

The mefenamic acid (synonym: 2-[(2,3-Dimethylphenyl)amino]benzoic acid, N-(2,3-Xylyl)anthranilic acid) was purchased from Sigma-Aldrich (Ref. No. M4267-50G). The CO_2_ gas (99.999% purity) was supplied by “Linde Gas Rus”.

### 2.2. Methods

#### 2.2.1. Quantum Chemical Calculations

The quantum chemical calculations (QCC) were performed in the Gaussian 16 program package [22]. The initial MA conformer structures were generated using the semi-empirical PM3 method, which was developed by J. J. P. Stewart and was first reported in 1989 [23]. The scanning was carried out when rotating the molecular fragments around four bonds in the MA molecule, which is schematically represented in Figure 1. The conformational search was based on the analysis of the potential energy surface scans. Thus, 18 conformers were found.

Then we optimized the MA molecule geometry by the DFT method with three functionals: (i) Becke’s hybrid three-parameter Lee-Yang-Parr functional, B3LYP [24,25], (ii) B3LYP with Grimme’s dispersion correction (GD3) [26], (iii) APFD [27] (which includes the Petersson–Frisch dispersion correction). The authors of the APFD functional note that it is suitable for conformational analysis, and our previous works show that it reproduces vibrational spectra with reasonable accuracy [9,10,11,12,28]. For all these functionals we used the 6–311++g(2d,2p) [29,30] basis set, which includes the polarization and diffuse functions necessary for correct H-bonding description. The structure of each conformer corresponded to the local potential energy minimum and was confirmed by frequency calculations. Finally, we obtained 16 MA conformers which had different structures and energies and no imaginary frequencies. The *Natural Bond Orbital* analysis (NBO) [31,32] and *Quantum Theory of Atoms In Molecule* (QTAIM) [33,34,35] methods were applied to study the intramolecular interactions. For the conformers found, we calculated the parameters of intramolecular hydrogen bonds and determined the energy barriers of transitions between these conformational states. We also calculated a set of IR vibrational bands for the most stable MA conformers in a vacuum in the wavenumber ranges corresponding to the IR spectroscopy experiment.

#### 2.2.2. Classical Molecular Dynamics Simulation

To analyze the conformational manifold of the MA molecules and the transitions between the conformation states in the scCO_2_ medium at different temperatures, we applied a metadynamic approach [36] within molecular dynamics simulation. The calculation was performed in the NPT ensemble at temperatures of 160, 190 and 220 °C using the GROMACS 4.5.5 program package. The simulation time was 10 ns. The CO_2_ solvent density was constant and corresponded to the isochore of 1.1*ρ*_cr._(CO_2_) chosen for the IR spectroscopy experiment. “Packmol” program package [37] was used to create the initial configuration of one MA molecule in 1024 CO_2_ molecules. We used Zhang’s [38] potential model for CO_2_, while the charges on the MA atoms were taken from the QCC results obtained using B3LYP–GD3/6–311++g(2d,p). The intermolecular interactions were described using the GAFF (General Amber Force Field) [39] and OPLS (Optimized molecular potential for liquid simulation) [40] force fields.

#### 2.2.3. In Situ IR Spectroscopy

To characterize the conformational equilibrium of MA molecules in scCO_2_ under isochoric heating conditions, we applied the experimental setup described in detail in our previous work [41]. Here, we also used a high-pressure–high-temperature (HPHT) optical cell with a variable optical path length [8] to measure the IR spectra. This cell with optical windows made of silicon makes it possible to record high resolution IR spectra in a wide range of thermodynamic parameters of state in the wavenumber range of 1000–7000 cm^−1^. To resist the high pressure and high temperature these windows are cylindrical in shape with an external diameter of 12 mm and are 9 mm thick, whereas their effective visible diameter is 8 mm. The experimental cell was heated by four cartridge heaters placed in the corners of its body. Three thermocouples were used for temperature control. One of them was in the vicinity of one of the four cartridge heaters and was connected to a proportional-integral-derivative (PID) controller allowing us to regulate the temperature with an accuracy of ±1 °C. The second thermocouple was placed in the upper part of the cell body and the third one was located in the bottom part of the cell. Such positioning of the thermocouples made it possible to control the temperature gradient in the internal volume of the cell. The necessary pressure inside the cell was created by injecting CO_2_. A manual pressure pump was used to manage the pressure value [41] with an accuracy of ±1 bar.

The following procedure was applied to prepare a one-phase binary mixture of MA–scCO_2_, which does not contain an excess of crystalline MA within the temperature range of interest. A small amount of the commercial crystalline MA powder, mainly comprised of polymorph I, was placed into the sample holder in the bottom part of the cell. The mass of this powder was sufficient to reach the total dissolution in scCO_2_ at the temperature of 130 °C. Then, we were absolutely sure that the binary MA–scCO_2_ mixture would not contain any excess crystalline MA in the system in the temperature range of 140–210 °C. Furthermore, to exclude the influence of the residual amounts of atmospheric components (especially, water and oxygen) on the sample, the HPHT cell was depressurized to a residual pressure of 10^–4^ bar. After that, the cell was filled with extra dry CO_2_ to reach the necessary pressure value corresponding to the chosen isochore of 1.1*ρ*_cr._(CO_2_) and preheated to 130 °C. To dry out the high-pressure CO_2_ we used a specially designed additional module included in the experimental setup. This module represented a thick-wall flow-type cylinder filled with zeolite and was connected to the output of the main high-pressure setup. Thus, when passing through this cylinder the high-pressure CO_2_ was additionally dried before it reached the HPHT cell.

The MA dissolution in scCO_2_ was monitored by analyzing the intensity change in some MA spectral bands. For this purpose, we performed successive registration of CO_2_ phase spectra for the binary mixture of MA–scCO_2_ at a temperature of 130 °C. When the intensity of the MA-related spectral bands stopped changing, we considered the dissolution process to be complete. Then, to prove that the MA concentration in scCO_2_ in the temperature range of 140–210 °C was constant, we performed the screening of the temperature dependence of the ν(C=O) spectral band integral intensity (see Appendix A).

In this study, the IR spectra of the binary MA–scCO_2_ mixture were recorded on a Bruker Vertex V80 FTIR spectrometer in the temperature range from 140 °C to 210 °C with a 10 °C step and a constant CO_2_ density corresponding to *ρ** = 1.1*ρ*_cr_(CO_2_), where *ρ*_cr_(CO_2_) = 10.625 mol·L^−1^ is the critical carbon dioxide density, the pressure for this isochore was in the range of 296–439 bar. The pressure values for each point of the phase diagram studied are presented in Table 1. To improve the quality of the measured spectra and reduce the influence of the atmospheric water vapor and carbon dioxide, we kept the working chamber of the spectrometer under vacuum during the experiment. For this purpose, we used a powerful oil-free vacuum pump to quickly reach the residual pressure of ~10^−4^ bar.

During the experiment, at each temperature the spectra were measured immediately after the target temperature was reached and then several times with a step of 15 min to obtain the time evolution of the spectra. The total measurement time at each temperature in the range of 140–180 °C was 6 h, at T = 190 °C it was 12 h, at T = 200 °C–40 h, and at 210 °C–12 h. To reduce the statistical error, the spectrum was recorded 128 times for each measurement and was then averaged out. The resolution of the obtained spectra was 1 cm^–1^. The resulting MA spectra in scCO_2_ were calculated by direct subtraction of the carbon dioxide spectra and those of the silicon windows from the initial (raw) spectra containing contributions of all the components: MA, CO_2_ and silicon windows, which were measured under the same thermodynamic conditions. Finally, the resulting spectra in the spectral range corresponding to the analytical spectral domain were corrected by baseline subtraction.

## 3. Results and Discussion

### 3.1. Conformational Analysis Quantum Chemical Calculations

The MA molecule structure is shown in Figure 1. It contains two groups, which can form intra- and intermolecular H-bonds. The MA molecule also has a dimethyl-substituted benzene ring and a carboxylic fragment, which is assumed to be responsible for its high conformational lability. There are four bonds in the MA molecule, around which the fragments can rotate, leading to the formation of various conformers. These bonds are numbered in Figure 1.

Based on the calculation results, we found that the most stable MA conformers are formed when the functional groups rotate around bonds “**1**” and “**2**”, while the rotation around bonds “**3**” and “**4**” leads to the formation of conformers with higher energies. Thus, within this work we considered only the four most stable conformers. They form when the carboxylic group rotates around bond “**1**” (variation of the dihedral angle **τ_1_** (C4–C3–C10–O11)) and the dimethyl-substituted benzene ring rotates around bond “**2**” (variation of the dihedral angle **τ_2_** (C4–N15–C17–C19)) in the MA molecule structure (see Figure 2). These conformers are denoted as Ia, Ib, IIa, IIb. The values of the mentioned dihedral angles for each of the four conformers are presented in Table 2. According to the energy values for these conformers (see Table 2), we divided them into two groups: Ia–Ib and IIa–IIb.

Then we calculated the energy barriers of the conformational transitions caused by varying the dihedral angles **τ_1_** and **τ_2_** for the three functionals used. These barriers are presented in Figure 3. Analyzing these plots, we found that the conformational transition barrier related to the variation of the dihedral angle **τ_1_** is high (~50 kJ·mol^−1^). However, the barrier of the conformational transition related to the variation of the dihedral angle **τ_2_** does not exceed 3 kJ·mol^−1^ (for B3LYP–GD3 and APFD) and 1.5 kJ·mol^−1^ (for B3LYP), which is lower than the thermal fluctuations (k_B_T) at the temperature of 130 °C. Thus, we determined the two most stable conformers Ia and Ib (see Figure 4), for which the difference in the energies and the energy barrier between them are insignificant. However, the higher barrier (~20 kJ·mol^−1^) is observed upon the transition through the plane of the other benzene ring and is related to the steric effects of the two methyl substituents. Nevertheless, the transition through this plane leads to the formation of mirror-symmetric conformers, which are indistinguishable in terms of energy. The same situation is observed for the second pair of conformers IIa and IIb (see Figure 4), which have higher energy values. Specifically, the transition between them can easily occur in the temperature range studied. However, as the insert presented in Figure 3 shows, the shape of the energy profile of the conformational transition related to the variation of the **τ_2_** dihedral angle, calculated using the B3LYP functional without a dispersion correction, differs from the shape of the profiles obtained using functionals with a dispersion correction. These differences are within the **τ_2_** region corresponding to the conformational transition (60°–140°). Thus, accounting for the dispersion correction somewhat changes the relative energies and, hence, reverses the conformer stability ranking within the Ia–Ib and IIa–IIb pairs. Therefore, for further analysis we analyzed only the results obtained using the B3LYP–GD3/6–311++g(2d,p) and APFD/6–311++g(2d,p) functionals.

The results presented above show that the main structural difference between the group of conformers Ia–Ib and IIa–IIb is in the carboxylic group orientation relative to the N–H group proton. The same division of the MA conformers into two groups was proposed in [42], but the authors also took into account the two conformers, formed upon the rotation around bond “**4**” and characterized by higher energy. These conformers were assigned to the first group. The transition between these two conformer groups occurs when the dihedral angle **τ_1_** changes by ~180°. For these conformers, we applied the NBO analysis to calculate the orbital stabilization energy (*E*^2^) of the electron pair acceptor (antibonding orbital BD*(N–H))*,* charge transfer (*q*) from the bonding orbital (LP(O)) to the antibonding orbital (BD*(N–H)). Using QTAIM analysis we calculated the electronic density (*ρ*) at the bond path critical point (3, −1), and the bond potential energy (½*V*). These calculations were performed for the interactions between the carbonyl oxygen and the proton of the N–H group (N–H…O=C) in conformers Ia and Ib, and between the N–H group proton and the hydroxyl oxygen (N–H…O_H_–C) in conformers IIa and IIb (see Table 3). According to the obtained results, the rotation related to variations in the **τ_1_** dihedral angle leads to the breakage of the strong intramolecular N–H…O=C hydrogen bond in conformers Ia and Ib, and to the formation of a weaker H-bond between the N–H group proton and the hydroxyl oxygen (N–H…O_H_–C) in conformers IIa and IIb.

As has been shown in our previous work [12], the value of ½*V* is in good correlation with *ρ*, and this was also shown in [43,44] for the case of strong and medium strength hydrogen bonds. However, the *E*^2^ value demonstrates a much worse correlation [12,43]. Moreover, as one can see from Table 3 for conformer Ia, the NBO analysis with the APFD functional gives overestimated values of *E*^2^ and *q*, as compared to those obtained using the B3LYP–GD3 functional. Therefore, in the present work we also used ½*V* as the energy measure of hydrogen bonding, as Espinosa et al. suggested [35]. For conformers Ia and Ib, the values of ½*V* lie in the range from –33.86 to –36.51 kJ·mol^−1^ (for the B3LYP–GD3 functional) and from –38.56 to –41.65 kJ·mol^−1^ (for the APFD functional). These values correspond to the energy of strong hydrogen bonds. Along with that, for these two functionals, these values for conformers IIa and IIb lie in the range from –28.13 to –29.85 kJ·mol^−1^ and from −29.97 to −31.97 kJ·mol^−1^, respectively. These values correspond to the threshold of strong hydrogen bonds and are in good agreement with the value *q*, which is close to 0.01 and is also close to the threshold of strong hydrogen bonds, according to Weinhold [31].

For two groups of conformers—Ia, Ib and IIa, IIb—we calculated a set of IR vibration frequencies, presented in Figure 5, within QCC using B3LYP–GD3 and APFD functionals. The presented spectra for each conformer, which were calculated using these two functionals, are in good qualitative agreement and can be used to analyze the experimental IR spectra. For convenience of presentation, each vibration frequency was described by a Lorentzian profile with the same dispersion. An analysis of this set of spectra allowed us to select two spectral domains lying in the wavenumber ranges of 1430–1610 cm^−1^ and 3300–3750 cm^−1^. These domains are bounded by dashed-line rectangles in Figure 5 and can be chosen as analytical ones when analyzing the experimental IR spectra. In the first domain (1430–1610 cm^–1^), the high-frequency contribution is associated only with the N–H group rocking vibration, whereas the low-frequency one contains several vibrational bands that are mainly related to the complex vibrations of the MA molecule aromatic system. The second wavenumber range (3300–3750 cm^−1^) contains only one spectral contribution related to the stretching vibration of the N–H functional group of the MA molecule.

The choice of these spectral domains as analytical ones was prompted by the fact that changing the MA conformations induces a direct change in the N–H vibrations because the amino-group is directly involved in the formation of an intramolecular hydrogen bond with the carbonyl or hydroxyl oxygen of the carboxylic fragment, depending on the orientation of the latter. Consequently, the change in the parameters of these H-bonds induced by the conformational transition directly affects the vibration parameters of the N–H group. On the other hand, the conformational transition can also indirectly influence the aromatic system vibration through electronic density redistribution. It occurs owing to its close positions to the MA functional groups (in particular, C=O, N–H, and O–H) that are sensitive to the conformation changes. Therefore, the appearance of a new spectral contribution in the considered domains, for instance, at a temperature increase, can be a signature of the new MA conformer emergence. In our recent works [6,7,8] we also showed the advantage of using the spectral domain associated with the aromatic system vibrations for screening conformational transitions of pharmaceutical molecules in the scCO_2_ phase.

Moreover, in our work [8] we demonstrated another reason for using the N–H stretching vibration mode as an analytical one. It is associated with the conformational type of MA polymorphism and with the difference in the N–H stretching band position for various MA polymorphs. The authors of [18,19] also showed that the transition from polymorph I to polymorph II shifts the position of this band from 3312 cm^−1^ to 3353 cm^−1^ and this shift can be considered a classical indicator of this polymorphic transition. Such changes indicate weakening of the intramolecular hydrogen bond, the formation of which involves the amino-group proton.

Additionally, analyzing the spectra presented in Figure 5 in the chosen analytical spectral domains, we can clearly divide conformers Ia, Ib, IIa and IIb into two groups. For the second group (IIa–IIb), the N–H stretching band is blue-shifted compared to that for the first one (Ia–Ib). On the contrary, in the spectral domain related to the N–H rocking vibration, the corresponding spectral band for the second group is slightly red-shifted relative to that for the first group. Finally, the complex spectral band for the second group, which is associated with the MA aromatic system vibration, is blue-shifted compared to that for the first group. Thus, this division of the conformers into two groups is in perfect agreement with the classification based on energetic and structural criteria presented above.

Finally, taking into account the probable dimerization of MA molecules in the solution, which involves the carbonyl oxygen and the hydroxyl group proton, we calculated the frequencies of the stretching vibration band of the C=O group for the dimers of conformers Ia and IIa. As this group is directly involved in the intermolecular H-bonding (see, e.g., [14,15]), these data allow us to determine the probability of dimer formation in the solution, based on the analysis of the ν(C=O) spectral band. Using the B3LYP–GD3 functional we found that this spectral band is centered at 1683 cm^–1^ for conformer Ia and at 1698 cm^–1^ for conformer IIa. According to Figure 5, their positions for monomeric conformers Ia and IIa are 1719 cm^−1^ and 1760 cm^−1^, respectively (such difference is related to the absence of the intramolecular H-bond, which involves the C=O group for conformer IIa). Thus, in the case of dimerization in the solution, where there is an equilibrium of two conformers, along with two ν(C=O) spectral contributions related to conformers Ia and IIa, in the experimental spectrum one can expect the appearance of at least one new spectral contribution red-shifted in relation to the spectral component of conformer Ia.

### 3.2. Conformational Analysis Molecular Dynamics Simulation

In this work, we chose GAFF and OPLS force fields to describe the intermolecular interactions in the binary MA–scCO_2_ mixture. This choice was motivated by satisfactory reproducibility of the potential energy curve obtained within QCC using the B3LYP–GD3 functional and associated with the conformational transition caused by variations in the τ_1_ dihedral angle (see Figure 6).

As has been shown above, according to QCC, due to the high energy barrier the probability of the transition between conformer pairs Ia–Ib and IIa–IIb, associated with the carboxylic group rotation, is low even at high temperatures. However, within QCC, the energy barriers as well as other energy parameters are calculated without taking into account the temperature and the medium influence. Accounting for these two factors becomes possible within molecular dynamics simulations. Thus, we performed MD simulations of MA in scCO_2_ at three temperatures: 160, 190 and 220 °C for the same isochore (see above), using the metadynamic approach. For each temperature, we calculated the 2D-surfaces of the Gibbs free energy in the coordinates of the same two dihedral angles (**τ_1_** and **τ_2_**) using two force fields (GAFF and OPLS). Then we obtained the corresponding 2D-surfaces of probability density based on the Boltzmann distribution and finally calculated the populations of conformers by integrating the corresponding peaks on the probability density map.

Figure 7 shows the 2D free energy maps obtained using the GAFF and OPLS force fields at 160 °C, as an example. The Gibbs free energy values of the two stable MA conformers and pass point height of the transition between them along with the τ_1_ and τ_2_ coordinates of these states are presented in Table 4. The normalized values of MA conformer populations are presented in Table 5. Comparing the values of the minimal potential barrier of the conformational transition presented in Table 4 with those obtained within QCC, we can conclude that the presence of scCO_2_ and temperature increase reduce the height of this barrier. This must increase the probability of a conformational transition. As one can see from the Gibbs free energy map, in contrast to QCC, there are only two free energy minima according to the MD simulation and, hence, there are only two conformers: Conf. I and Conf. II. Consequently, these pairs can be further considered as two conformers denoted as Conf. I and Conf. II.

### 3.3. Conformational Analysis IR Spectroscopy

The obtained IR spectra of MA dissolved in scCO_2_ in the temperature range of 140–210 °C along the 1.1*ρ*_cr._(CO_2_) isochore are presented in Figure 8 for two spectral domains—1000–2000 cm^−1^ and 3250–3500 cm^−1^. For each temperature we showed an equilibrium spectrum at the target temperature (see the experimental details in Section 2.2.3). First, an analysis of the ν(C=O) spectral band allowed us to exclude the dimerization of the molecules of MA in its solution in scCO_2_ over the whole temperature range studied. Indeed, according to QCC data, we could expect the emergence of a low-frequency spectral band shifted by 21–36 cm^−1^ in relation to that of conformer Ia centered at 1709 cm^−1^ (see Figure 8). However, as it can be seen from this figure, such spectral band is not observed, whereas the spectral band related to conformer Ia remains almost symmetric up to 190 °C. This was also shown in our previous work [8], where we studied a saturated MA solution in scCO_2_, where the MA concentration was a lot higher. Moreover, the abrupt redistribution of the ν(C=O) spectral bands intensities, related to conformers Ia and IIa, in the temperature range of 180–200 °C corresponds to the conformational transition. In contrast, for thermal destruction of intermolecular H-bonds we can expect a gradual redistribution of these band intensities, as has been shown for liquid water in one of our works [45].

Relying on the quantum chemical calculations and MD simulation results, we chose two spectral domains, which will be analyzed below. These areas are bounded by dashed-line rectangles in Figure 8. A qualitative analysis of the spectral changes in the wavenumber range of 3250–3500 cm^−1^, which is related to the N–H stretching vibrations, showed that in the temperature interval of 140–190 °C, there are two spectral contributions centered at 3344 cm^−1^ and 3430 cm^−1^. According to our QCC, the first one, which is dominant, can be assigned to MA Conf. I. Consequently, the second one, which has much lower intensity, can be assigned to MA Conf. II. Then, starting from 190 °C the intensity of the low-frequency component considerably decreases. Along with this, the intensity of the high-frequency component becomes higher. Finally, in the temperature range of 200–210 °C it becomes dominant. However, because the MA concentration in the CO_2_ phase is low and, consequently, the intensities of these two spectral contributions (as well as the signal-to-noise ratio) over the whole temperature range are low too, it is not rational to use this spectral domain for the quantitative analysis of the conformational equilibrium. Nevertheless, the general trend of the spectral changes is in good agreement with that of the changes in the spectral domain related to the C=O stretching vibrations (see above), as well as with the results of our previous work, where we studied the conformational equilibrium of MA molecules in the scCO_2_ phase contacting with solid MA.

Thus, for quantitative analysis of the experimental IR spectra we took the wavenumber range of 1420–1560 cm^−1^. We used the standard procedure of spectral band deconvolution into spectral components. According to our QCC, to reproduce the analytical spectral domain we applied a model that included two spectral contributions for each of the two MA conformers (see Figure 9a). For each of the conformers, the high-frequency contribution is related to the N–H rocking vibration and the low-frequency one is associated with the complex vibrations of the MA molecule aromatic system. Within our model, we used Lorentzian profiles to approximate the spectral contributions. This choice was prompted by the fact that MA molecules exist in the CO_2_ phase as monomers and the probability of the formation of hydrogen bonds between them is extremely low due to a very low concentration (~10^−4^ mole fractions). Along with that, to improve the approximation quality and to properly reproduce the experimental spectra within this spectral domain, we also introduced additional spectral profiles necessary to fit the spectral bands neighboring the analytical ones (see Figure 9a). We used Pseudo-Voigt profiles for their approximation. To realize the fitting procedure, we applied “Fityk” freeware [46] that includes a spectrum fitting tool. Figure 9a shows, as an example, a typical deconvolution of an experimental spectrum in the wavenumber range of 1400–1800 cm^−1^ including the analytical spectral domain of 1420–1560 cm^−1^ for the temperature of 140 °C. The quality of deconvolution and reproducibility of the experimental spectra in the wavenumber range corresponding to the analytical spectral domain for some of the studied temperatures are demonstrated in the Appendix A.

As a result of the spectra deconvolution, we obtained a set of parameters for each spectral component constituting the analytical spectral domain. Figure 9b–d show the temperature dependences of peak positions (ν_max_), dispersions of peaks or its half width at half height (γ) and integral intensity (A_int_), respectively. As one can see from Figure 9b, the temperature increase does not lead to a strong shift in the maximum positions of these spectral profiles. Even the maximum shift value does not exceed 6 cm^–1^, which can be linked only to the thermal fluctuation of vibration frequencies. However, the temperature dependencies of dispersions of these spectral profiles behave differently (see Figure 9c). Indeed, the dispersions of the spectral bands, which are attributed to the vibration of the aromatic system of Conf. I and Conf. II, considerably increase upon heating, whereas the dispersions of the spectral bands assigned to the N–H rocking vibration slightly decrease. Moreover, major changes take place in the temperature range of 190–200 °C. It is only in the case of the spectral profile attributed to the complex vibrations of the aromatic system of Conf. II that the major dispersion change takes place in the temperature range of 140–190 °C. Then this value remains almost constant.

The temperature dependencies of the integral intensities of the spectral components lying in the analytical spectral domain show the opposite behavior for conformers I and II (see Figure 9d). Taking into account that each of the conformers is responsible for two spectral contributions to the analytical spectral domain (see above) and assuming that the extinction coefficients of these bands are similar, we summed the integral intensities of these spectral components related to each conformer by pairs. The temperature dependencies of the obtained values (A′) are presented in Figure 10a. It is obvious that in the temperature range of 140–180 °C, the A′ value of the spectral components related to Conf. I is dominant. Along with this, the A′ values for both Conf. I and Conf. II remain almost constant. In the temperature range from 180 °C to 200 °C, abrupt redistribution of the A′ values for Conf. I and Conf. II is observed. The same behavior is also observed for the N–H and C=O stretching vibration bands. Finally, the A′ value that is related to Conf. II becomes dominant.

Based on the temperature dependences of A′ for the two conformers, we calculated their mole fractions (X*_i_*) in scCO_2_ vs. temperature as $X_{i}(T)={A^{\prime }}_{i}(T)/\sum_{i}{A^{\prime }}_{i}(T)$. The X*_i_*(*T*) dependencies are presented in Figure 10b. As one can see from this figure, in the temperature range of 140–180 °C both MA conformers exist in scCO_2_, and Conf. I prevails (76–73% depending on temperature). In the temperature range of 180–200 °C abrupt redistribution of these conformer mole fractions is observed. In addition, finally, in the temperature range of 200–210 °C, Conf. II prevails (77–82%). Thus, we do not observe a total conformational transition of the MA molecules in scCO_2_. However, in our previous work [8], we showed that in the case of a heterogeneous MA–scCO_2_ binary mixture, which has an excess of the MA crystalline phase, at temperatures below 160 °C, the content of Conf. II in scCO_2_ is negligible and is associated with the presence of a small amount of MA polymorph II in the initial MA crystalline phase. Moreover, the total conformational transition of the MA molecules occurs in the scCO_2_ phase already in the temperature range of 180–190 °C leading to the disappearance of Conf. I.

We observed a similar phenomenon in a series of our previous works, in which we studied the conformational equilibria of pharmaceuticals in the scCO_2_ phase contacting with an excessive amount of these drugs. In the case when the scCO_2_ phase is in contact with the drug crystalline phase [6,8,11], the full conformational transition of pharmaceutical molecules occurring in the scCO_2_ phase is related to the polymorphic transition in the crystalline drug phase induced by the temperature increase. On the contrary, when the scCO_2_ phase is in contact with the drug that is in the amorphous state [7,12], in the scCO_2_ phase, the temperature does not lead to a full conformational transition of the pharmaceutical molecules. Thus, when solid drugs are in contact with the scCO_2_ phase, their polymorphs determine the conformational equilibrium of the molecules of those drugs in the scCO_2_ phase. However, when pharmaceuticals, which are in contact with the scCO_2_ phase, are amorphous, a thermodynamic conformational equilibrium of drug molecules is realized in the solution. Similar behavior was observed in the present work for an unsaturated MA solution in scCO_2_, where the conformational equilibrium of MA molecules is determined only by the thermodynamic state of the system.

## 4. Conclusions

In our previous work [8], we found that if there is an interface between the crystalline MA and MA solution in scCO_2_, the conformational transition in the saturated solution phase is directly related to the polymorphic transition in the crystalline MA phase of this system. Moreover, the full polymorphic transition of MA from form I to form II that occurs at high temperatures results in degeneration of conformer I in the saturated solution phase. As it follows from [14], under ambient conditions, the order of stability of MA polymorphs is I > II > III; the enantiotropic transformation that takes place at higher temperatures changes this order to II > I > III. However, according to QCC also presented in [8], we showed that the potential energy of conformer II is higher than that of conformer I. Namely all these findings determined the aim of the present work—to study the molecular mechanism of conformational transitions of MA molecules.

For this purpose, we studied the conformational equilibria of MA molecules in a true solution of MA in CO_2_ under isochoric heating conditions in the temperature range of 140–210 °C. We found that in the considered phase diagram region, the equilibrium of two conformers is always realized in solution. In the temperature range of 140–180 °C, conformer I prevails (76–73% depending on temperature). In the temperature range of 180–200 °C a drastic redistribution of conformer I and II populations is observed. Finally, in the temperature range of 200–210 °C conformer II becomes dominant (77–82%). Therefore, according to the conformer distribution as a temperature function (obtained in the present work), it is obvious that the free energy of conformer II at the temperatures above 200 °C becomes lower than that of conformer I and vice versa at low temperatures. Taking into account that the potential energy of conformer II is higher than that of conformer I, as follows from QCC, this crossover can be classified as an entropy-driven phenomenon.

This fact is of great importance to pharmacy, for instance, when RESS is used to produce micronized forms of drug compounds with a desired polymorphic modification. In particular, if the target polymorph is not stable, the target micronized substance must not contain an admixture of other more stable polymorphic forms. To achieve this, it is important to maintain the interface between the crystalline API phase and the saturated fluid solution. Otherwise, crystallization from a true API solution will lead to the formation of a polymorph mixture even at the temperature, at which the conformer related to the target polymorph prevails. Such mixture may contain a side polymorph, along with the prevailing one. It is extremely undesirable, especially when the side polymorph stability is higher than that of the target polymorph.

## Figures and Tables

**Figure 1 materials-16-01403-f001:**
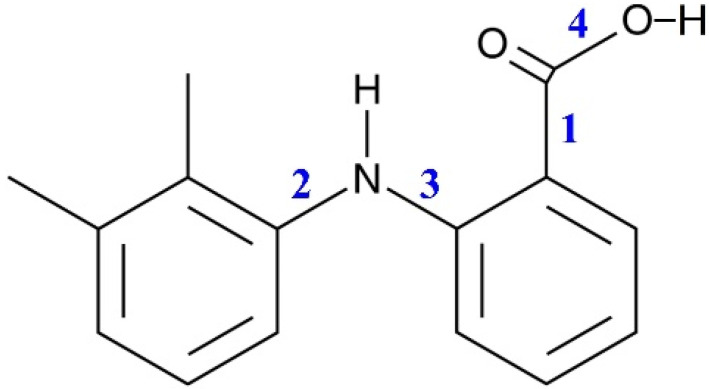
Schematic representation of the MA molecule structure. The numbers indicate the bonds around which the fragments can rotate, leading to the formation of various conformers.

**Figure 2 materials-16-01403-f002:**
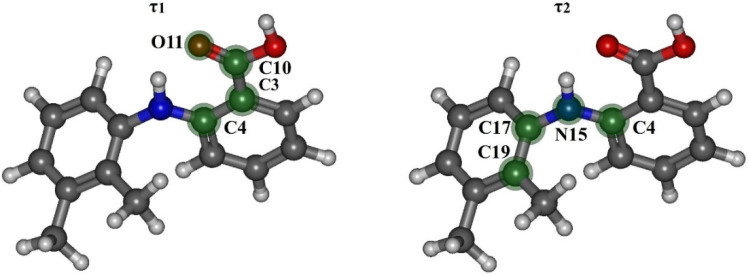
Schematic representation of the MA molecule with atoms forming dihedral angles **τ_1_** (C4–C3–C10–O11) and **τ_2_** (C4–N15–C17–C19). These atoms are shown in green color.

**Figure 3 materials-16-01403-f003:**
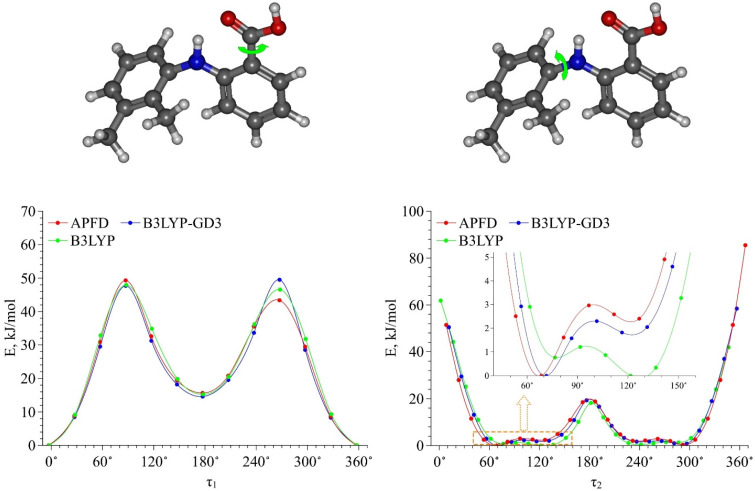
Energy barriers of conformational transitions related to variations of the dihedral angles **τ_1_** and **τ_2_** determined using three functionals. The insert on the plot of the energy profile of conformational transition related to variations of the dihedral **τ_2_** angle shows the difference in the energy profiles in the angle range of 60°–140°.

**Figure 4 materials-16-01403-f004:**
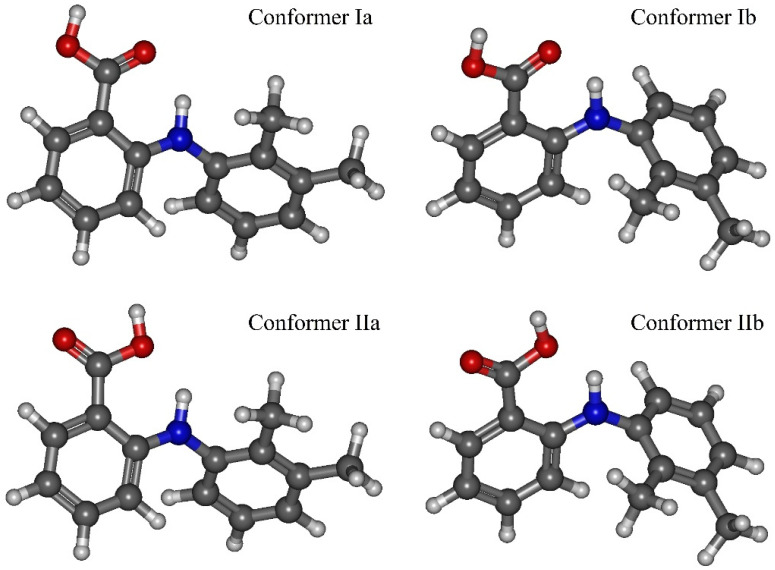
Schematic representation of the MA molecule conformers: Ia, Ib, IIa, IIb.

**Figure 5 materials-16-01403-f005:**
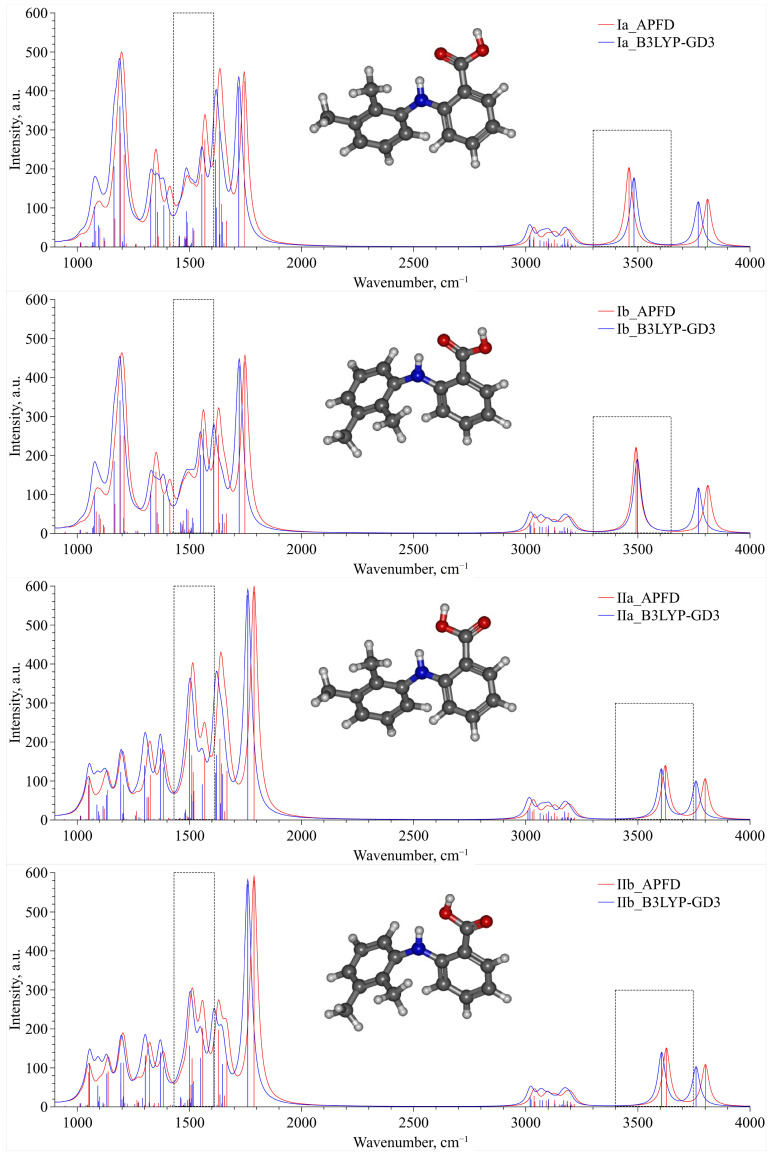
IR spectra for the four most stable conformers obtained within QCC using the B3LYP–GD3 and APFD functionals. The area bounded by dashed-line rectangles corresponds to the analytical spectral domains.

**Figure 6 materials-16-01403-f006:**
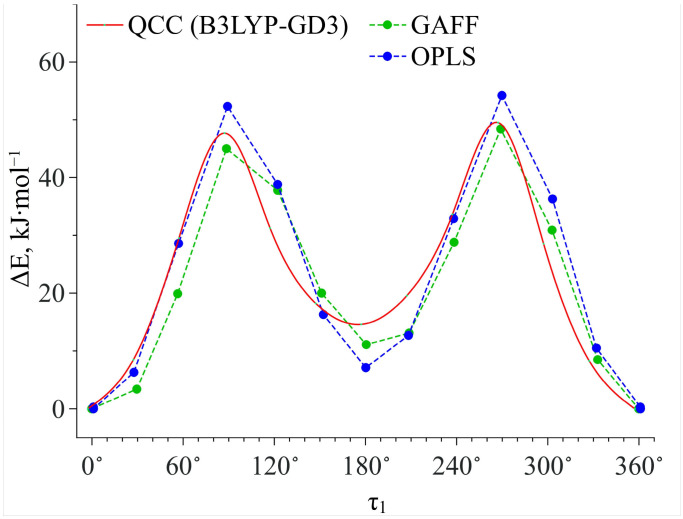
Comparison of the potential energy profiles along the dihedral angle **τ_1_** obtained within QCC using the B3LYP–GD3 functional and those obtained with the GAFF and OPLS force fields.

**Figure 7 materials-16-01403-f007:**
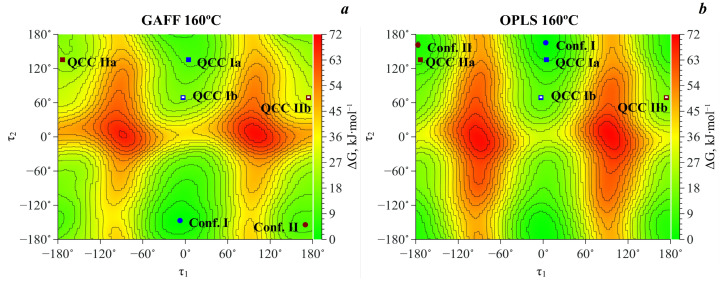
2D Gibbs free energy maps in the coordinates of two dihedral angles **τ_1_** and **τ_2_** obtained using GAFF (**a**) and OPLS (**b**) force fields at 160 °C for the MA molecule in scCO_2_. The symbols on the energy map show the MA conformer positions obtained from MD simulations (**Conf. I** and **Conf. II**)—circles, and from QCC (**QCC Ia**, **QCC Ib** and **QCC IIa**, **QCC IIb**)—squares.

**Figure 8 materials-16-01403-f008:**
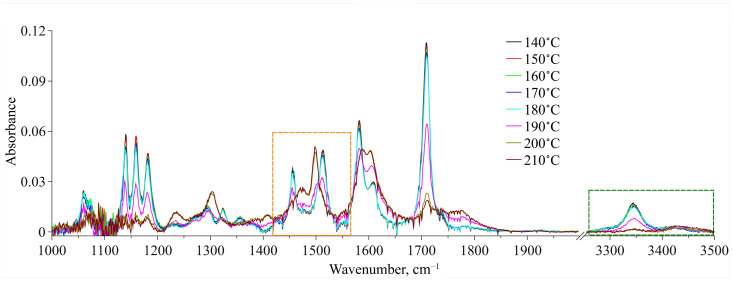
IR spectra of MA dissolved in scCO_2_ under isochoric heating conditions. The areas bounded by rectangles correspond to the analytical spectral domains.

**Figure 9 materials-16-01403-f009:**
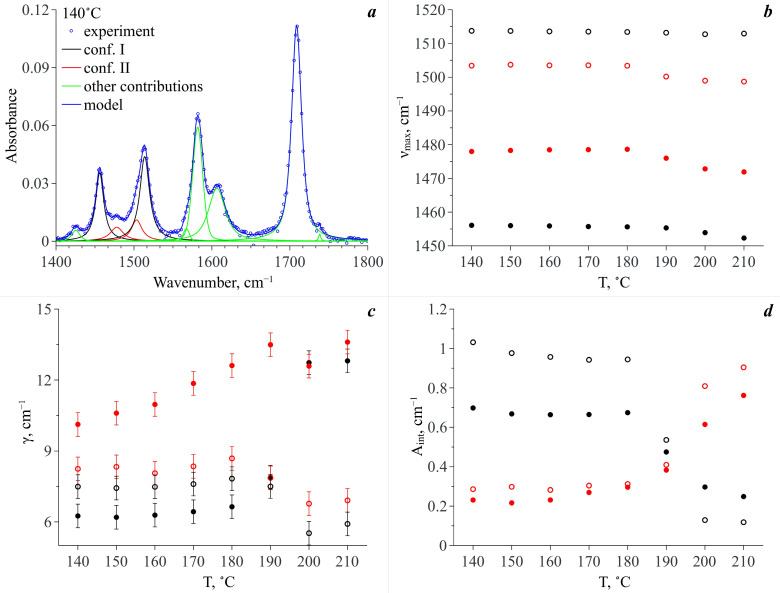
(**a**) An example of the experimental spectrum deconvolution for the temperature of 140 °C. Temperature dependences of: (**b**) the maximum positions of the spectral profiles (**ν_max_**); (**c**) dispersions of the spectral profiles (**γ**); (**d**) integral intensities of the spectral profiles (**A_int_**). Hereinafter the black circles correspond to MA Conf. I and the red circles denote MA Conf. II; the solid circles correspond to the spectral bands, which are attributed to the complex vibrations of the aromatic system and the open circles denote the spectral bands assigned to the N–H group rocking vibrations.

**Figure 10 materials-16-01403-f010:**
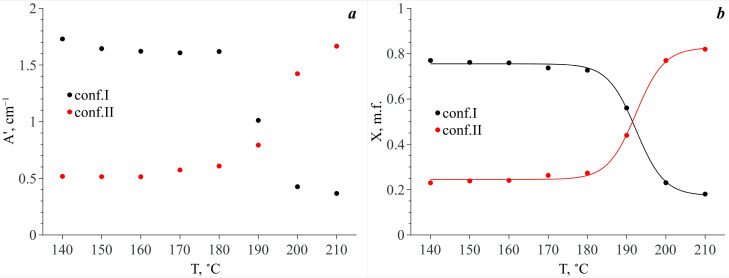
(**a**) Temperature dependencies of the integral intensity values summed pairwise (A′) for the spectral profiles that correspond to Conf. I and Conf. II (see Figure 9); (**b**) Temperature dependences of the mole fractions (X) of these conformers calculated as $X_{i}={A^{\prime }}_{i}/\sum_{i}{A^{\prime }}_{i}$.

**Table 1 materials-16-01403-t001:** Pressure values corresponding to the temperatures lying on the 1.1*ρ*_cr._(CO_2_) isochore.

T, °C	P, bar
140.00	296.03
150.00	316.58
160.00	337.09
170.00	357.57
180.00	378.02
190.00	398.43
200.00	418.80
210.00	439.13

**Table 2 materials-16-01403-t002:** Values of the dihedral angles **τ_1_** and **τ_2_** of the four most stable conformers that were obtained by QCC applying three functionals.

Functional	Dihedral Angle	Conf. Ia	Conf. Ib	Conf. IIa	Conf. IIb
B3LYP	**τ_1_**	4.09°	−2.17°	−173.95°	175.27°
	**τ_2_**	135.10°	76.64°	135.65°	77.39°
	∆E, kJ·mol^−1^	0.00	1.48	16.05	16.66
B3LYP–GD3	**τ_1_**	4.13°	−2.77°	−173.14°	174.12°
	**τ_2_**	132.06°	71.47°	132.16°	72.39°
	∆E, kJ·mol^−1^	1.14	0.00	16.65	14.56
APFD	**τ_1_**	4.34°	−2.87°	−173.04°	174.23°
	**τ_2_**	135.44°	68.48°	135.82°	69.36°
	∆E, kJ·mol^−1^	1.45	0.00	18.07	15.63

**Table 3 materials-16-01403-t003:** Orbital stabilization energy *E^2^* of the electron pair acceptor (antibonding orbital BD*(N–H))*,* charge transfer *q* from the bonding orbital (LP(O)) to the antibonding acceptor orbital (BD*(N–H)) (calculated within NBO), electron density *ρ* at the (3, −1) bond path critical point, and bond potential energy ½*V* (calculated within QTAIM) for MA conformers.

Functional	Conformer	*E*^2^, kJ·mol^−1^	*q*,e	*ρ*_(N−H…O)_, a.u.	−½*V*_(N−H…O)_, kJ·mol^−1^
B3LYP–GD3	Ia	42.17	0.0219	0.03360	36.50758
	Ib	39.37	0.0202	0.03183	33.85582
	IIa	28.49	0.0099	0.02835	29.85194
	IIb	22.72	0.0081	0.02712	28.13223
APFD	Ia	72.51	0.0412	0.03676	41.65356
	Ib	46.28	0.0224	0.03479	38.55547
	IIa	26.78	0.0092	0.02966	31.96542
	IIb	24.18	0.0082	0.02828	29.97008

**Table 4 materials-16-01403-t004:** Gibbs free energy values of two MA conformers and minimal energy barriers of the transition between them in scCO_2_ calculated using GAFF and OPLS force fields at three temperatures.

	T = 160 °C			T = 190 °C			T = 220 °C		
	Conf. I	Barrier	Conf. II	Conf. I	Barrier	Conf. II	Conf. I	Barrier	Conf. II
GAFF									
**τ_1_**, °	−7.3	−102.9	168.9	14.7	102.9	−161.6	14.7	102.9	−157.9
**τ_2_**, °	−146.9	−139.6	−154.3	150.6	161.6	154.3	154.3	150.6	150.6
ΔG, kJ·mol^−1^	0.0	38.16	10.25	0.0	36.99	9.89	0.0	34.36	8.05
OPLS									
**τ_1_**, °	3.7	−95.5	−176.3	0.0	95.5	−172.6	3.7	−91.8	−180.0
**τ_2_**, °	165.3	−154.3	161.6	−161.6	172.7	154.3	172.7	−180.0	165.3
ΔG, kJ·mol^−1^	0.0	47.54	3.88	0.0	45.31	3.95	0.0	44.74	2.51

**Table 5 materials-16-01403-t005:** Normalized MA conformer populations obtained using GAFF and OPLS force fields.

	GAFF			OPLS		
	T = 160 °C	T = 190 °C	T = 220 °C	T = 160 °C	T = 190 °C	T = 220 °C
Conf. I	0.9451	0.9288	0.8769	0.7428	0.7284	0.6919
Conf. II	0.0463	0.0375	0.0732	0.2572	0.2716	0.3081

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
