# Peer review of "Molecular Mechanism of Conformational Crossover of Mefenamic Acid Molecules in scCO2"

_materials, 2023, doi:10.3390/ma16041403_

Round 1
Reviewer 1 Report
This is a brilliant paper devoted to the study of the conformational equilibria of mephenamic acid in scCO2 under isochoric heating conditions. Using experimental methods (infrared spectroscopy) and molecular modeling (quantum chemical calculations), it was clearly proven which conformers of mephenamic acid predominate in scCO2 solutions as a function of temperature. These results are of great interest for pharmacy, especially for the production of micronized forms of drug compounds with the desired polymorphic modification. Thus, this manuscript can be recommended for publication in the journal "Materials" without any changes.
Author Response
Dear reviewer, we are grateful for the high mark you gave to our manuscript and recommending it for publication in the journal "Materials".
Reviewer 2 Report
The paper presents interesting results, which is of interest, the literature review is very comprehensive it is reasonably well written and of interest of the journal. However it needs minor corrections before publication, as follows:
1- In abstract, add a sentence that explains your choice of range 140−210°Ð¡
2- At the end of your introduction, you did not specify the novelty of your work compared to the works cited in literature;
3- In figure 6 where (a) and (b)
4- In figure 8 where (a), (b), (c) and (d)
5- In figure 9 where (a) and (b)
6- Standardize the references, as an example the ref [25-28], you have ignored the DOI, the ref [32] you have ignored the numbers of pages and volume?
Reviewer 3 Report
This paper studied the conformational equilibria of mefenamic acid by quantum-chemical calculation, MD simulation in scCO2 and experimental analysis by FT-IR. I think that the contents are meaningful in this area and the paper is written well. There are a few minor points for correction as follows.
* Structure of Ia, Ib, IIa, IIb
The authors explain the structure of Ia, Ib, IIa and IIb in line 229-252. I think the structure will help the reader to understand the explanation. On the other hand, the structure is shown in Figure 4. I think that these structures should be shown in the explanation at line 229-252.
*Figure 8
There is no (a), (b), (c), (d) in Figures. Please check.
*Figure 9
There is no (a), (b) in Figures. Please check.
Reviewer 4 Report
The paper „Molecular mechanism of conformational crossover of mefenamic acid molecules in scCO2” (matrials-2146321) is devoted to conformational equilibria of mefenamic acid in solution in scCO2 under isochoric heating. The paper could be published but significant changes of the manuscript are necessary.
1. For the reader convenience it should be explained why the very specific conditions for studying conformational changes were chosen (scCO2 solution, isochoric heating).
2. Authors takes into account only two phases of the mefenamic acid what gives an impression that only these two phases exist. It seems necessary to include a review of all sold state structures of mefenamic acid (Acta Cryst. E, 2001, 63, o1656; Cryst. Str. Comm. 1976, 5, 861; J. Pharm. Sci. 2004, 93, 144; Pharm. Res. 2006, 23, 2375, Cryst. Growth. Des, 2012, 12, 4285; Cryst. Growth. Des, 2012, 12, 5521;Pharm 2017, 9, 16) with detailed description of the structural differences connected with the rotational polymorphism. Furthermore, there is a discrepancy between of the existence of only two phases of fenamic acid the mentioned on page 2 with the phase transition between phases I, II, and III mentioned at the same page.
3. The main point not taken into account is formation of the dimer of mefenamic acid. Carboxylic groups of two mefenamic acid molecules are connected by strong OHO hydrogen bond that are stable because of the OHO hydrogen bond strength and symmetry of the formed dimer. This fact determines rotation of other bonds so the theoretical calculation should be repeated for the mefenamic acid dimer.
4. Also interpretation of the IR spectra should be done taking into account not only the intramolecular NHO hydrogen bond but also formation of the OHO dimers of mefenamic acid. Assignment of the IR bands should be in agreement with previous interpretation of experimental IR spectra of mefenamic acid (Spectrochimica Acta Part A: Molecular and Biomolecular Spectroscopy 96, 2012, 972–985; J. Pharm. Biomed. Anal. 37, 2005, 509–515; J. Pharm. Biomed. Anal. 149, 2018, 603–611).
5. In the calculation description please indicate which program was used to perform the QTAIM analysis.
6. The language should be corrected to avoid slang and shortcuts.
Round 2
Reviewer 4 Report
In my opinion the manuscript has been sufficiently improved to warrant publication in Materials.